# MoCLIP: Motion-Aware Fine-Tuning and Distillation of CLIP for Human Motion Generation

## Abstract

*Human motion generation is essential for fields such as animation, robotics, and virtual reality, requiring models that effectively capture motion dynamics from text descriptions. Existing approaches often rely on Contrastive Language-Image Pretraining (CLIP)-based text encoders, but their training on text-image pairs constrains their ability to understand temporal and kinematic structures inherent in motion and motion generation. This work introduces MoCLIP, a fine-tuned CLIP model with an additional motion encoding head, trained on motion sequences using contrastive learning and tethering loss. By explicitly incorporating motion-aware representations, MoCLIP enhances motion fidelity while remaining compatible with existing CLIP-based pipelines and seamlessly integrating into various CLIP-based methods. Experiments demonstrate that MoCLIP improves Top-1, Top-2, and Top-3 accuracy while maintaining competitive FID, leading to improved text-to-motion alignment results. These results highlight MoCLIP's versatility and effectiveness, establishing it as a robust framework for enhancing motion generation.*

## 1. Introduction

Generating realistic human motion is a challenging goal in computer vision and graphics, with broad applications in fields such as animation [1, 4, 7, 51], virtual/augmented reality [10, 14], gaming [20, 46, 54], and robotics [40, 44]. Human motion generation remains challenging due to the high diversity of possible movements [5]. Models must learn complex spatio-temporal dynamics and generate physically plausible, meaningful sequences. Moreover, collecting large-scale datasets with rich annotations is difficult [27], and it often requires advanced frameworks for automated annotating process [26]. Motion capture data is costly to acquire and often limited in semantic scope [42]. Even recent datasets that pair motions with textual labels cover only a portion of the motion manifold and fail to capture the full richness of natural language descriptions [42].

A variety of methods have been explored to tackle human motion generation. Conditional generation is a common theme, where motion is produced in response to some input like an action category, a textual description, a past motion sequence, or pose [13, 29, 52]. More recent approaches incorporate natural language as a conditioning signal, aiming to generate motions from text descriptions. These text-to-motion models demonstrated encouraging results on short, template-like descriptions (e.g. "a person walks forward") and small datasets, but often struggle with longer or more complex descriptions that go beyond the training data's distribution [29, 48].

To overcome data limitations, researchers use generative frameworks and pre-trained models. Diffusion models, successful in image and audio generation, have been adapted for motion generation, producing smooth and diverse motions with State-Of-The-Art (SOTA) performance [9, 43]. However, while diffusion-based methods continue to advance, they remain computationally demanding.

CLIP-based approaches leverage the rich prior knowledge from contrastive language–image pre-training (CLIP) [35] to enhance motion models with semantic understanding. By aligning motion representations with CLIP's vision-language feature space, these methods benefit from the broad semantic coverage learned from 400 million image-text pairs. MotionCLIP [42] pioneered this alignment by training a motion autoencoder whose latent space corresponds directly to CLIP's text and image embeddings, enabling motion synthesis from novel textual prompts without modifying CLIP's pre-trained representations.

While these approaches enhance motion generation by leveraging CLIP's semantic structure, CLIP itself is primarily trained on text-image pairs and is not explicitly tailored for capturing temporal progression or intricate kinematic details. Although it effectively models relationships between language and static visual content, applying CLIP-based representations directly to motion tasks may not fully account for the temporal coherence and natural movement patterns required for high-fidelity motion generation.

In this work, we propose MoCLIP, a novel human motion generation model that explicitly extends the standard CLIP architecture by integrating a dedicated motion encoder trained via contrastive learning on motion sequences. Unlike MotionCLIP, which maintains CLIP's pre-trained embeddings, MoCLIP fine-tunes CLIP's text encoder to shift its embeddings toward motion-oriented representations, inherently capturing the temporal dynamics and intricate kinematic details essential for realistic motion synthesis. Additionally, MoCLIP incorporates a distillation mechanism (tethering loss) to preserve CLIP's rich semantic knowledge while adapting it explicitly to the motion domain. By constructing a joint motion-text latent space, MoCLIP aligns motion sequences with corresponding natural language descriptions, enabling our transformer-based motion generator to produce semantically coherent, high-fidelity human motion.

Our model maintains compatibility with existing CLIP-based pipelines, allowing seamless integration into any system. By systematically exposing CLIP's encoder to motion-sequence data, MoCLIP refines the alignment between textual prompts and 3D motion representations without sacrificing the model's broad language-understanding capabilities. Quantitatively, MoCLIP achieves superior or competitive results against SOTA, and qualitatively it exhibits robust generalization to novel inputs.

Our research explores the following contributions:

- Enhancing CLIP for human motion by introducing an additional motion head trained with contrastive learning, augmenting a standard CLIP encoder to encode temporal and kinematic aspects of motion data into the textual latent space.
- We demonstrate MoCLIP's effectiveness in capturing motion dynamics by integrating it into three different vision pipelines, improving Top-1, Top-2, and Top-3 accuracy over standard CLIP-based models.
- To analyze the effects of our training and contributions, we conduct an ablation study, comparing an advanced MoCLIP with a naive baseline by examining different training features.

## 2. Related Works

Early works primarily used recurrent neural networks (RNNs) for deterministic sequence generation. Approaches such as [2, 3, 19] employed RNN-based sequence-to-sequence models to map text directly to motion. However, these models suffered from limited diversity and struggled to maintain temporal coherence in longer sequences.

**Variational and Autoencoder-based Models:** Variational Autoencoders (VAEs) are being utilized in many computer vision fields [25, 39, 53]. As for motion generation, models such as T2M [15] and TEMOS [29] introduced transformer-based VAEs to learn shared latent spaces, enabling diverse motion generation conditioned on text descriptions. These approaches improved text-motion alignment and diversity metrics over earlier deterministic methods [14, 16, 29]. Furthermore, reciprocal generation approaches such as TM2T [16] demonstrated improved semantic alignment through simultaneous training of text-to-motion and motion-to-text tasks.

**Autoregressive Transformer Models:** recently gained popularity [11, 28, 45]. In motion generation, leveraging discrete latent representations obtained via vector quantized variational autoencoders (VQ-VAEs) [5, 12, 37] has become a known practice. Notably, methods such as T2M-GPT [47], MMM [31], and BAMM [30] demonstrated significant performance gains in realism, diversity, and semantic alignment by modeling human motion generation as discrete token prediction. T2M-GPT, for example, combined a simple CNN-based VQ-VAE with GPT-style transformers, achieving top-level fidelity scores (low FID) and competitive semantic alignment [47]. MMM and BAMM introduced masked autoregressive strategies to handle bidirectional context effectively, improving controllability and sequence coherence [30, 31]. Thus VQ-VAEs, have become foundational components in contemporary motion synthesis, facilitating discrete latent representation, which significantly improved diversity and semantic alignment with text compared to previous approaches [16, 30, 31, 47].

**CLIP-based Approaches:** Recent methods have utilized pre-trained vision-language models such as CLIP [34] for human motion generation. MotionCLIP [42] aligns motion latent spaces directly with CLIP's semantic text embeddings, enabling zero-shot generalization. However, CLIP embeddings primarily capture static image-text semantics, which may not fully represent temporal and kinematic details essential for realistic motion synthesis. Additionally, direct fine-tuning of CLIP embeddings can lead to catastrophic forgetting of pre-trained semantic knowledge [24]. MoCLIP addresses these issues with a specialized fine-tuning strategy that explicitly integrates motion-aware representations into CLIP's semantic space without requiring a separate motion encoder during inference, thus effectively balancing semantic retention and temporal specificity.

## 3. Methodology

We propose a specialized fine-tuning strategy for the CLIP model to capture the spatio-temporal patterns inherent in graph-based human motion data. To achieve this, we fine-tune the textual embeddings using a distillation loss, which constrains the adaptation process, ensuring the embeddings retain the core semantics of the original CLIP representation

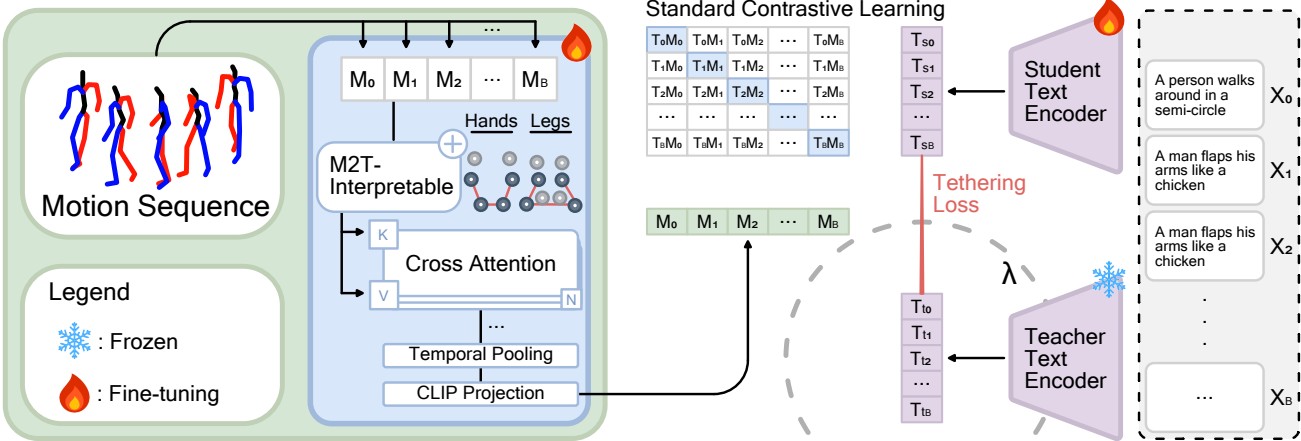

Figure 1. Overview of our MoCLIP training framework, which fine-tunes CLIP for human motion representation. We adopt M2T-Interpretable [36] as the motion encoder to extract spatio-temporal embeddings from a motion sequence. This encoder includes cross-limb attention to capture fine-grained inter-limb coordination. The resulting motion embeddings $M = \{M_0, M_1, ..., M_B\}$ are aligned with text embeddings via a contrastive loss. To preserve CLIP's broad semantic knowledge, we introduce a distillation loss (Tethering Loss), which constrains the student text encoder using the pre-trained teacher text encoder. The parameter $\lambda$ controls the influence of this constraint, balancing adaptation and semantic retention.

while incorporating motion-specific details. Additionally, we employ a cosine similarity loss to guide the embeddings toward preserving the directional continuity of motion, reinforcing alignment between the text and motion spaces.

M2T-Interpretable [36] introduced a guided motion encoding framework that currently represents the state-of-the-art in human motion captioning, demonstrating its ability to map human motion into a rich and semantically meaningful latent space. This method captures both fine-grained motion details and high-level semantic structures. Given its performance in aligning motion with textual descriptions, we selected this approach as the foundation for our motion encoder. Its ability to generate robust motion embeddings with strong semantic coherence makes it the best candidate for converting motion data into CLIP's latent space.

A novel extension in our method is the introduction of cross-limb attention connections that extend beyond conventional skeletal adjacency constraints to [36]. Specifically, we introduce direct attention connections between both hands and both feet, allowing the model to better capture inter-limb coordination in complex human actions, such as clapping, jumping, or balancing. These additional attention pathways improve the model's ability to recognize non-local interactions, as recognized in [8, 22, 33, 38].

Finally, temporal attention mechanisms are applied to the encoded motion features before pooling along the temporal dimension. The resulting spatio-temporal representations are projected into CLIP's pre-trained multimodal embeddings. This structured approach preserves CLIP's broad semantic knowledge while introducing a motion-aware adaptation.

## 3.1. Loss Functions & Optimization

We employ a multi-term loss function to achieve effective contrastive alignment, preserve original semantic representations via feature-space distillation (tethering), and explicitly align motion embeddings with textual semantics.

The primary objective is to align motion embeddings $z_{\text{motion}}$ with their corresponding text embeddings $z_{\text{text}}$. We use a symmetric cross-entropy loss following standard contrastive learning practice:

$$\mathcal{L}_{\text{contrastive}} = \frac{1}{2} \left[ \text{CE}(z_{\text{motion}} W z_{\text{text}}^\top, y) + \text{CE}(z_{\text{text}} W z_{\text{motion}}^\top, y) \right] \tag{1}$$

where $z_{\text{motion}}, z_{\text{text}} \in \mathbb{R}^{N \times d}$ are normalized embeddings from the motion encoder and CLIP text encoder, respectively; $W \in \mathbb{R}^{d \times d}$ is a learnable scaling matrix derived from CLIP's logit scaling parameter; and $y \in \{1, \ldots, N\}$ are the ground-truth matching indices for the $N$ sample pairs in a mini-batch. CE denotes cross-entropy loss, defined as:

$$\text{CE}(X, y) = -\frac{1}{N} \sum_{i=1}^{N} \log \left( \frac{\exp(X_{i,y_i})}{\sum_{j=1}^{N} \exp(X_{i,j})} \right) \tag{2}$$

**Feature Distillation Loss (Tethering Loss)** Inspired by recent works in CLIP fine-tuning, such as CLIP-CITE [21] and LDIFS [24], we introduce a feature distillation loss. This regularization loss ensures that fine-tuned CLIP text embeddings remain close to their original pre-trained representations, thus mitigating catastrophic forgetting.

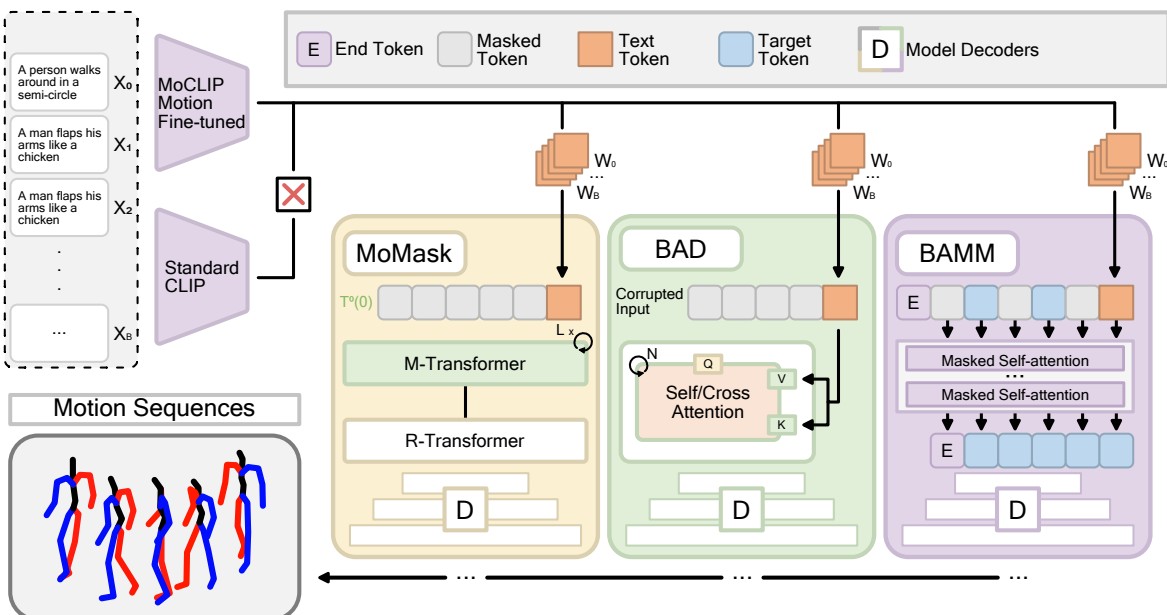

Figure 2. Example integration of the MoCLIP model into existing text-to-motion generation pipelines. MoCLIP serves as a direct replacement for the standard CLIP encoder previously utilized in various established models, including MoMask [17], BAD [18], and BAMM [30]. The figure illustrates simplified architectures of these models, demonstrating how MoCLIP seamlessly substitutes standard CLIP by providing motion-aware text embeddings.

We employ the L2 loss between the fine-tuned CLIP embeddings (student) $z_{\text{text-student}}$ and the original pre-trained CLIP embeddings (teacher) $z_{\text{text-teacher}}$:

$$\mathcal{L}_{\text{distill}} = \frac{1}{N} \sum_{i=1}^{N} \left\| z_{\text{text-student}}^{(i)} - z_{\text{text-teacher}}^{(i)} \right\|_2^2 \qquad (3)$$

We specifically select MSE for the distillation loss because it permits the student embeddings to shift in a manner not strictly constrained by angular (cosine) similarity to the teacher embeddings. This flexibility allows the student embeddings to more effectively align with the motion embeddings, while still remaining within a controlled proximity to the original CLIP embedding space.

The strength of the distillation term is controlled by a hyperparameter $\lambda_{\text{distill}}$. We empirically determine the optimal value of $\lambda_{\text{distill}}$ through a comprehensive ablation study, analyzing its impact on embedding alignment and fine-tuning performance per model.

**Motion-Text Cosine Alignment Loss**   To explicitly align the semantic directionality of motion embeddings toward text embeddings, we introduce an additional cosine-similarity alignment loss. This encourages the fine-tuned embeddings not only to match contrastively but to closely reflect motion-specific semantic relationships directly:

$$\mathcal{L}_{\text{alignment}} = 1 - \frac{1}{N} \sum_{i=1}^{N} \frac{z_{\text{motion}}^{(i)} \cdot z_{\text{text-student}}^{(i)}}{\|z_{\text{motion}}^{(i)}\|_2 \|z_{\text{text-student}}^{(i)}\|_2} \qquad (4)$$

This loss term specifically encourages a direct semantic alignment between text and motion representations beyond the standard contrastive pairing.

Our total optimization objective is the weighted sum of the three described loss terms:

$$\mathcal{L}_{\text{total}} = \mathcal{L}_{\text{contrastive}} + \lambda_{\text{distill}} \mathcal{L}_{\text{distill}} + \mathcal{L}_{\text{alignment}} \qquad (5)$$

This multi-objective approach aims for embedding alignment while preserving crucial pre-trained knowledge from the original CLIP model.

## 4. Experiments

### 4.1. Datasets

Two major datasets are generally used for motion generation tasks, namely HumanML3D [14] and KIT-ML [32]. The proposed model relies on pre-trained weights from each chosen baseline model on these datasets. Because pre-trained models for KIT-ML were unavailable, only HumanML3D was used to demonstrate the benefits of MoCLIP.

| Methods | R Precision↑ | | | FID↓ | MM-Dist↓ | Diversity↑ | MModality↑ |
|---|---|---|---|---|---|---|---|
| | Top 1 | Top 2 | Top 3 | | | | |
| TM2T [16] | $0.424^{\pm0.003}$ | $0.618^{\pm0.003}$ | $0.729^{\pm0.002}$ | $1.501^{\pm0.017}$ | $3.467^{\pm0.011}$ | $8.589^{\pm0.076}$ | $2.424^{\pm0.093}$ |
| T2M [14] | $0.455^{\pm0.003}$ | $0.636^{\pm0.003}$ | $0.736^{\pm0.002}$ | $1.087^{\pm0.021}$ | $3.347^{\pm0.008}$ | $9.175^{\pm0.083}$ | $2.219^{\pm0.074}$ |
| MDM [43] | - | - | $0.611^{\pm0.007}$ | $0.544^{\pm0.044}$ | $5.566^{\pm0.027}$ | $9.559^{\pm0.086}$ | $2.799^{\pm0.072}$ |
| MLD [6] | $0.481^{\pm0.003}$ | $0.673^{\pm0.003}$ | $0.772^{\pm0.002}$ | $0.473^{\pm0.013}$ | $3.196^{\pm0.010}$ | $9.724^{\pm0.082}$ | $2.413^{\pm0.079}$ |
| MotionDiffuse [49] | $0.491^{\pm0.001}$ | $0.681^{\pm0.001}$ | $0.782^{\pm0.001}$ | $0.630^{\pm0.005}$ | $3.113^{\pm0.001}$ | $9.410^{\pm0.049}$ | $1.553^{\pm0.042}$ |
| T2M-GPT [48] | $0.492^{\pm0.001}$ | $0.679^{\pm0.002}$ | $0.775^{\pm0.002}$ | $0.141^{\pm0.004}$ | $3.121^{\pm0.009}$ | $9.761^{\pm0.081}$ | $1.831^{\pm0.048}$ |
| ReMoDiffuse [50] | $0.510^{\pm0.005}$ | $0.698^{\pm0.002}$ | $0.795^{\pm0.004}$ | $0.103^{\pm0.004}$ | $2.974^{\pm0.016}$ | $9.018^{\pm0.075}$ | $1.795^{\pm0.043}$ |
| MMM [31] | $0.504^{\pm0.003}$ | $0.696^{\pm0.003}$ | $0.794^{\pm0.002}$ | $0.080^{\pm0.003}$ | $2.998^{\pm0.007}$ | $9.577^{\pm0.050}$ | $1.164^{\pm0.041}$ |
| MoMask [17] | $0.521^{\pm0.002}$ | $0.713^{\pm0.002}$ | $0.807^{\pm0.002}$ | $\mathbf{0.045}^{\pm0.002}$ | $2.958^{\pm0.008}$ | - | $1.241^{\pm0.040}$ |
| MoMask+MoCLIP | $\mathbf{0.533}^{\pm0.002}$ | $\mathbf{0.730}^{\pm0.002}$ | $\mathbf{0.823}^{\pm0.001}$ | $0.047^{\pm0.002}$ | $\mathbf{2.868}^{\pm0.006}$ | $9.619^{\pm0.082}$ | $\mathbf{1.242}^{\pm0.040}$ |
| BAMM [30] | $0.522^{\pm0.003}$ | $0.715^{\pm0.003}$ | $0.808^{\pm0.003}$ | $\mathbf{0.055}^{\pm0.002}$ | $2.936^{\pm0.077}$ | $9.636^{\pm0.009}$ | $1.732^{\pm0.051}$ |
| BAMM+MoCLIP | $\mathbf{0.531}^{\pm0.003}$ | $\mathbf{0.724}^{\pm0.003}$ | $\mathbf{0.819}^{\pm0.002}$ | $0.064^{\pm0.003}$ | $\mathbf{2.871}^{\pm0.008}$ | $\mathbf{9.713}^{\pm0.083}$ | $\mathbf{1.749}^{\pm0.054}$ |
| BAD [18] | $\mathbf{0.517}^{\pm0.002}$ | $\mathbf{0.713}^{\pm0.003}$ | $\mathbf{0.808}^{\pm0.003}$ | $0.065^{\pm0.003}$ | $\mathbf{2.901}^{\pm0.008}$ | $\mathbf{9.694}^{\pm0.068}$ | $\mathbf{1.194}^{\pm0.044}$ |
| BAD+MoCLIP | $0.510^{\pm0.003}$ | $0.706^{\pm0.002}$ | $0.801^{\pm0.002}$ | $\mathbf{0.062}^{\pm0.003}$ | $2.941^{\pm0.008}$ | $9.613^{\pm0.076}$ | $1.152^{\pm0.044}$ |

Table 1. Quantitative results comparing MoCLIP-integrated models (MoMask, BAMM, and BAD) against baseline methods on the HumanML3D dataset. Metrics include R-Precision (Top-1, Top-2, Top-3), Frechet Inception Distance (FID), Multimodal Distance (MultiDist), Diversity, and Multimodality. Arrows (↑,↓) indicate whether higher or lower values represent better performance, respectively. Best results for each method are bolded.

HumanML3D is a large-scale 3D motion-language dataset for motion understanding and generation, built from AMASS [23] and HumanAct12 [13] motion data. It contains 14,616 motion sequences with 44,970 textual descriptions, covering diverse activities like daily actions, sports, and acrobatics. Each motion lasts 2–10 seconds, downsampled to 20 FPS, totaling 28.59 hours. To enhance diversity, mirrored motions with adjusted descriptions are included. The dataset supports text-to-motion synthesis, motion retrieval, and activity recognition. The dataset is split into 80% training, 15% validation, and 5% test sets, following the standard setup used in previous works.

### 4.2. Metrics

**R-Precision** Evaluates motion-to-text retrieval accuracy by ranking the Euclidean distances between a given motion sequence and 32 text descriptions (1 ground-truth and 31 mismatched). The retrieval performance is measured using Top-1, Top-2, and Top-3 accuracy, indicating the proportion of cases where the correct text description appears within the top-ranked results.

**Frechet Inception Distance (FID)** is a widely used metric for evaluating the quality of generated images by comparing their statistical similarity to real images. It computes the Fréchet distance between feature representations of real and generated images extracted from a pre-trained Inception v3 [41] network.

**Multimodal Distance** is the average Euclidean distance between features from different modalities, such as text and generated motion, measured in a shared latent space to evaluate cross-modal alignment quality.

**Multimodality** measures the diversity of generated motions from a single text description. For each description, 20 motion sequences are generated, forming 10 motion pairs. The Euclidean distances between motion features in each pair are computed, and the final score is the average distance across all text descriptions, reflecting the model's ability to generate diverse outputs from the same input.

### 4.3. Experimental Setup

This paper investigated the effectiveness of MoCLIP embeddings in VQ-VAE-based methods. To validate this, we selected the top three transformer-based VQ-VAE approaches to implement our MoCLIP model. We evaluated across these motion generation frameworks, including MoMask [17], BAMM [30], and BAD [18], integrating it into their architectures without modifying their core designs.

We begin by loading a pre-trained CLIP model as the student network, modifying it to accept our motion encoder. A teacher CLIP model is also loaded and kept frozen throughout training. To ensure a stable adaptation process, the student CLIP text encoder is initially frozen for 35 epochs, allowing the motion encoder to align with the pre-trained CLIP embeddings. After this phase, the student text encoder is unfrozen and trained alongside the motion encoder using the loss functions described above for an additional 15 epochs for a total of 50 epochs.

All models are trained using their pre-trained weights, including VQ-VAE weights. We fine-tune them with MoCLIP at a lower learning rate of 1e-6 and without a warm-

up phase. For three-stage training models like MoMask and BAMM, both the T2M Transformer and Residual Transformer are trained separately with MoCLIP to align their learned embeddings with the modified CLIP space. All models are trained for 200 epochs on HumanML3D using AdamW as the optimizer, maintaining the original hyperparameters used in their respective training pipelines. Training is conducted on A6000 GPUs.

## 5. Results

**R-Precision Improvement** MoCLIP improves retrieval accuracy across multiple models. In MoMask, MoCLIP increases Top-1 R-Precision from 0.521 to 0.533 (+1.2%), Top-2 from 0.713 to 0.730 (+1.7%), and Top-3 from 0.807 to 0.823 (+1.6%). A similar trend is observed in BAMM, where Top-1 R-Precision rises from 0.522 to 0.531 (+0.9%), Top-2 from 0.715 to 0.724 (+0.9%), and Top-3 from 0.808 to 0.819 (+1.1%). These improvements demonstrate a consistent enhancement in motion-text alignment across retrieval-based models.

**Motion Quality: FID and Multimodal Distance** MoCLIP maintains perceptual quality while improving motion-text consistency. For MoMask, FID increases slightly from 0.045 to 0.047 (+0.002), while Multimodal Distance decreases from 2.958 to 2.868 (-0.09). This suggests a trade-off where improved alignment comes with a minor increase in perceptual difference. In BAMM, FID increases from 0.055 to 0.064 (+0.009), whereas Multimodal Distance decreases from 2.936 to 2.871 (-0.065). While FID increases slightly.

**Performance Drop in BAD** Unlike MoMask and BAMM, the BAD model does not benefit from MoCLIP integration, exhibiting a slight decrease in retrieval accuracy. While The FID score improves from 0.065 to 0.062 (-0.003), Top-1 R-Precision moves from 0.517 to 0.510 (-0.7%), Top-2 from 0.713 to 0.706 (-0.7%), and Top-3 from 0.808 to 0.801 (-0.9%). Additionally, Multimodal Distance increases from 2.901 to 2.941 (+0.04), suggesting a weaker motion-text relationship.

This degradation is likely due to the underlying architecture of BAD. Unlike token-based generative models, BAD employs Bidirectional Autoregressive Diffusion, which combines sequential and bidirectional attention through a permutation-based corruption technique. While this enables BAD to effectively capture long-range motion dependencies, it may also make the model more sensitive to modifications in its embedding space. MoCLIP may introduce subtle shifts in BAD's learned dependencies, leading to weaker retrieval accuracy. This suggests that BAD's autoregressive models with bidirectional constraints might not integrate as

effectively with MoCLIP as with other models.

MoCLIP improved retrieval accuracy and motion-text consistency in MoMask and BAMM, yielding a 1.2–1.7% increase in R-Precision and a 2–3% reduction in Multimodal Distance. However, its integration into BAD leads to minor performance drops, likely due to architectural incompatibilities. These findings suggest that MoCLIP is more effective in token-based models, whereas bidirectional autoregressive architectures may require additional adaptation to fully leverage its benefits.

## 6. Ablation Study

We conducted comprehensive ablation studies to evaluate the effectiveness and importance of individual components within MoCLIP, using our foundational baselines. Specifically, we examine two types of training: a naïve baseline, employing basic contrastive learning without specialized positional encodings or targeted attention mechanisms, and an advanced version, incorporating positional encodings, targeted attention toward critical body parts, tethering loss and cosine similarity alignment. Both studies aim to quantify the impacts of these training variations and feature enhancements on human motion generation performance.

### 6.1. Impact of Tethering Loss

To determine the optimal balance between retaining CLIP's semantic knowledge and adapting to motion-specific tasks, we thoroughly examined the influence of the tethering loss weight ($\lambda$). We selected multiple candidate values, specifically $\lambda \in 0, 0.2, 0.4, 0.6, 0.8, 1.0$, for all three baseline models. During these experiments, we maintained a consistent experimental setup: each model was trained for a total of 50 epochs using a combination of contrastive learning, cosine alignment, and tethering loss. We monitored performance metrics such as Frechet Inception Distance (FID) and Multimodal Distance (MM-Dist) at regular intervals to capture nuanced shifts in performance as $\lambda$ varied.

### 6.2. Naïve Model Training Comparison

We further explored the necessity and efficacy of our specialized fine-tuning and additional architectural enhancements (positional encodings and targeted attention mechanisms). To this end, we developed a naïve baseline model that utilized only basic contrastive learning without the specialized positional encoding or enhanced attention toward critical body parts such as hands and feet. To evaluate the effect of embedding fine-tuning schedules, we experimented by unfreezing text embeddings for varied periods—specifically for the last 2, 5, 7, and 10 epochs—allowing us to gauge the impact of different fine-tuning durations.

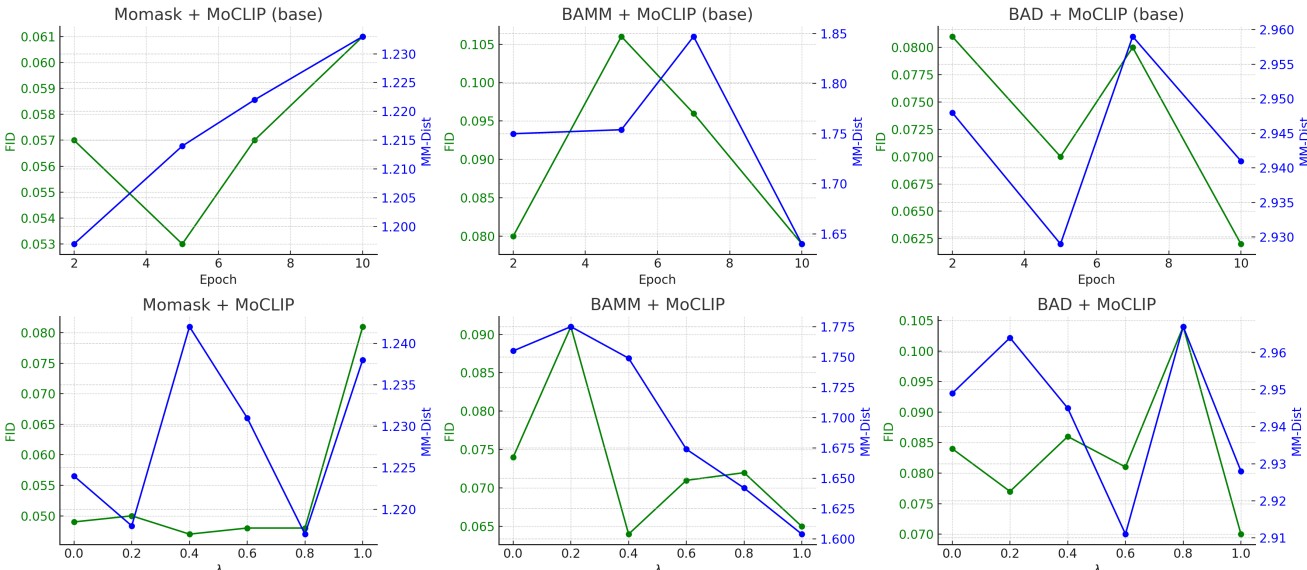

Figure 3. Ablation studies examining the impact of fine-tuning duration in naive training (top row) and tethering loss weight $\lambda$ (bottom row) on model performance, measured by Frechet Inception Distance (FID, green axis) and Multimodal Distance (MM-Dist, blue axis). Each plot compares different fine-tuning epochs (2, 5, 7, and 10 epochs) for naive baseline training (top) and varying tethering strengths ($\lambda$ from 0.0 to 1.0) for advanced MoCLIP model and training (bottom). Lower values indicate improved perceptual quality (FID) and better multimodal alignment (MM-Dist). Results are averaged over multiple runs on the HumanML3D dataset.

## 6.3. Experimental Setup

All models in the ablation were evaluated under identical conditions using HumanML3D. Each model configuration was trained and then tested twenty times to ensure the reliability and statistical significance of the reported results.

## 6.4. Analysis and Findings

In Table 2 and Figure 3, we present detailed results from our ablation study, evaluating the impact of varying fine-tuning epochs and tethering loss weights ($\lambda$) on naive model of MoCLIP (top rows). MoMask achieves optimal FID (0.053) at 5 epochs, balancing performance with retrieval accuracy (Top-1: 0.538), while additional epochs enhance accuracy but negatively impact FID. BAMM achieves its best overall naive MoCLIP performance at 10 epochs, presenting the lowest FID (0.079) and simultaneously the highest retrieval accuracy (Top-1: 0.541). Notably, BAD uniquely benefits from extended naive training compared to the advanced model and training, steadily improving across metrics and achieving the best naive FID (0.062) at 10 epochs. Given this performance relative to advanced methods, the naive-trained BAD model was selected for final use.

In contrast, for the advanced models trained with tethering methods (bottom rows), model selection prioritized optimal FID along with consistency across metrics. MoMask demonstrated its strongest performance at a moderate tethering weight of $\lambda = 0.4$, achieving the best overall FID (0.047), accompanied by robust retrieval accuracy (Top-1: 0.533) and stable performance across MM-Dist and diversity metrics. Similarly, BAMM attained its lowest FID (0.064) and consistently balanced performance metrics at $\lambda = 0.4$, supporting this choice for final deployment. However, advanced training approaches for BAD did not show significant metric improvements over naive training, prompting the selection of the naive-trained model at 10 epochs for final implementation.

## 7. Conclusion

We introduced MoCLIP, a straightforward to implement fine-tuning strategy that directly substitutes the standard CLIP encoder with minimal adjustments. MoCLIP aligns CLIP's text embeddings with motion-aware representations through contrastive learning, a tethering loss to preserve semantic consistency, and a cosine similarity alignment loss to semantically align motion-text embeddings. Experiments demonstrated consistent improvements in semantic alignment and retrieval accuracy, with MoMask improving Top-1 R-Precision from 0.521 to 0.533 (+1.2%) and BAMM improving from 0.522 to 0.531 (+0.9%), while maintaining competitive FID scores (MoMask: from 0.045 to 0.047; BAMM: from 0.055 to 0.064). MoCLIP provides immediate performance gains at a low implementation cost.

However, our results indicate that some model architectures may not benefit equally from this fine-tuning method.

| Methods | R Precision↑ | | | FID↓ | MM-Dist↓ | Diversity↑ |
|---|---|---|---|---|---|---|
| | Top 1 | Top 2 | Top 3 | | | |
| MoMask (2 epochs) | 0.537 | _0.733_ | _0.826_ | 0.057 | _2.838_ | _9.667_ |
| MoMask (5 epochs) | 0.538 | 0.729 | 0.822 | 0.053 | 2.856 | **9.703** |
| MoMask (7 epochs) | _0.539_ | 0.730 | 0.824 | 0.057 | 2.859 | 9.623 |
| MoMask (10 epochs) | **0.540** | **0.736** | **0.829** | 0.061 | **2.826** | 9.654 |
| MoMask ($\lambda = 0.0$) | 0.532 | 0.727 | 0.822 | 0.049 | 2.875 | 9.645 |
| MoMask ($\lambda = 0.2$) | 0.536 | 0.730 | 0.824 | 0.050 | 2.864 | 9.643 |
| MoMask ($\lambda = 0.4$) | 0.533 | 0.730 | 0.823 | **0.047** | 2.868 | 9.619 |
| MoMask ($\lambda = 0.6$) | 0.534 | 0.731 | 0.823 | _0.048_ | 2.862 | 9.614 |
| MoMask ($\lambda = 0.8$) | 0.536 | 0.732 | 0.824 | _0.048_ | 2.864 | 9.654 |
| MoMask ($\lambda = 1.0$) | 0.529 | 0.724 | 0.818 | 0.081 | 2.905 | 9.645 |
| BAD (2 epochs) | 0.512 | 0.704 | 0.801 | 0.081 | 2.948 | 9.533 |
| BAD (5 epochs) | **0.518** | **0.710** | **0.805** | 0.070 | 2.929 | 9.578 |
| BAD (7 epochs) | 0.511 | 0.704 | 0.799 | 0.080 | 2.959 | 9.584 |
| BAD (10 epochs) | 0.510 | 0.706 | 0.801 | **0.062** | 2.941 | _9.614_ |
| BAD ($\lambda = 0.0$) | 0.511 | 0.705 | 0.800 | 0.084 | 2.949 | 9.539 |
| BAD ($\lambda = 0.2$) | 0.512 | 0.701 | 0.797 | 0.077 | 2.964 | 9.609 |
| BAD ($\lambda = 0.4$) | 0.511 | 0.704 | 0.799 | 0.086 | 2.945 | 9.555 |
| BAD ($\lambda = 0.6$) | _0.516_ | 0.709 | _0.806_ | 0.081 | **2.911** | 9.574 |
| BAD ($\lambda = 0.8$) | 0.507 | 0.699 | 0.796 | 0.104 | 2.967 | **9.617** |
| BAD ($\lambda = 1.0$) | 0.517 | _0.709_ | 0.805 | _0.070_ | _2.928_ | 9.577 |
| BAMM (2 epochs) | 0.531 | _0.733_ | _0.829_ | 0.086 | _2.839_ | _9.825_ |
| BAMM (5 epochs) | 0.526 | 0.724 | 0.819 | 0.106 | 2.894 | **9.832** |
| BAMM (7 epochs) | 0.515 | 0.712 | 0.808 | 0.096 | 2.960 | 9.808 |
| BAMM (10 epochs) | **0.541** | **0.740** | **0.835** | 0.079 | **2.805** | 9.759 |
| BAMM ($\lambda = 0.0$) | 0.528 | 0.723 | 0.818 | 0.074 | 2.878 | 9.665 |
| BAMM ($\lambda = 0.2$) | 0.527 | 0.720 | 0.814 | 0.091 | 2.917 | 9.734 |
| BAMM ($\lambda = 0.4$) | 0.530 | 0.724 | 0.819 | **0.064** | 2.871 | 9.713 |
| BAMM ($\lambda = 0.6$) | 0.534 | 0.728 | 0.824 | _0.071_ | 2.855 | 9.721 |
| BAMM ($\lambda = 0.8$) | _0.535_ | 0.729 | 0.824 | 0.072 | 2.849 | 9.746 |
| BAMM ($\lambda = 1.0$) | 0.520 | 0.712 | 0.808 | 0.065 | 2.917 | 9.669 |

Table 2. Ablation study results for MoMask, BAD, and BAMM with various naive fine-tuning epochs vs tethering loss strengths ($\lambda$). Best scores are bolded, second is underlined.

For instance, BAD experienced a slight decrease in Top-1 R-Precision (from 0.517 to 0.510) and an increase in Multimodal Distance (from 2.901 to 2.941), suggesting that some architectures may require targeted fine-tuning approaches or architectural refinements to fully leverage these embeddings.

In future work, we aim to further validate MoCLIP's effectiveness by expanding evaluation across additional motion generation architectures such as diffusion models, as well as more datasets suchs as KIT-ML. Additionally, exploring architecture-specific fine-tuning strategies and investigating adaptive fine-tuning techniques for individual models may yield further improvements in performance and generalization.

# Acknowledgment

The acknowledgement has been omitted in compliance with CVPR's double-blind policy.

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
