# OpenReview forum: "MoCLIP Motion-Aware Fine-Tuning and Distillation of CLIP for Human Motion Generation"
_thecvf.com/CVPR/2025/Workshop/HuMoGen — CVPR 2025 Workshop HuMoGen Submission_

### Official Review · Reviewer_VTwV · 2025-03-24
**The motivation is interesting, the writing needs polishing and method part lacks clarity.**

**Rating:** 3
**Confidence:** 4

**Review:**

This paper proposes MoCLIP, an extension of the standard CLIP text encoder that shifts embeddings toward motion-aware representations. To retain CLIP's prior knowledge, a distillation scheme is introduced. While the motivation is interesting, the writing needs polishing, and the explanation of the implementation procedure and method lacks clarity.

Based on the provided scripts, I have the following comments and questions:

- Is there any reference for the tethering loss mentioned in the paper? Based on Eq. 3, the applied distillation strategy seems to follow the basic approach. Providing a reference or more details on the tethering loss would help readers understand its novelty and how it improves upon existing distillation strategies.

- What is the difference between the $\mathcal{L}_{\text{contrastive}}$ and $\mathcal{L}_{\text{alignment}}$ loss functions? Both are used to align the motion and text embeddings, but their specific roles and differences are not clearly explained. A more detailed explanation of how these two loss functions differ in their approach to alignment would help clarify their individual contributions and the rationale for using both.

- The training procedure is not sufficiently explained. The paper introduces the training pipeline in lines 317-337, but the statement **"both T2M Transformer and residual transformers are trained *separately with* MoCLIP"** (line 331) is unclear. Does this mean MoCLIP is fine-tuned with the parameters in **MoMask** and **BAMM**, or do the parameters of MoMask and BAMM undergo further fine-tuning based on a pre-fine-tuned MoCLIP text encoder with fixed parameters?  Additionally, **Figure 2** is intended to illustrate how MoCLIP works, but it is not clear enough to demonstrate how MoCLIP is utilized in the training process. More clarification is needed in both the text and figures.

- Is there any experiment or feature visualization demonstrating the difference between embeddings extracted by the original CLIP text encoder and the fine-tuned version? In other words, how can the authors prove that the text embeddings produced by **MoMask** are more motion-aware compared to those from the vanilla CLIP model? Including such an analysis or visualization would strengthen the paper and validate the claimed improvements.

---

### Official Review · Reviewer_yujz · 2025-03-26

**Rating:** 4
**Confidence:** 4

**Review:**

# Paper Summary
The paper proposes a novel motion generation framework, which incoporating the cross-attention mechanism between different body parts and the Tethering loss to transfer clip's knowledge to the motion domain. The framework achieves the SOTA result and can incoporate with other motion generation framework.
# Paper Strengths
1. The paper proposes a cross-attention mechanism between different body parts, which empirically can achieve a better motion embedding, and the ablation study proves that.
2. The paper proposes a Tethering loss to distill the original clip knowledge into the motion domain, which lead to a more precise motion-related text embedding.
3. The framwork can incoporate with other motion generation frameworks and can lead to an improvement, which is meaningful.
# Paper Weaknesses
1. As you train the motion and text embedding using contrastive loss, so beyond motion generation, is it possible that your framework can achieve SOTA result in motion retrieval task compared to TMR[1] and LAVIMO[2]?

[1] Petrovich, Mathis, Michael J. Black, and Gül Varol. "Tmr: Text-to-motion retrieval using contrastive 3d human motion synthesis." Proceedings of the IEEE/CVF International Conference on Computer Vision. 2023.

[2] Yin, Kangning, et al. "Tri-modal motion retrieval by learning a joint embedding space." Proceedings of the IEEE/CVF Conference on Computer Vision and Pattern Recognition. 2024.

---

### Meta-Review · Area_Chair_7sm4 · 2025-03-29

**Recommendation:** Accept

**Metareview:**

The reviewers have divergent assessments of this paper. Reviewer yujz rates it as a weak accept (4/5) with high confidence (4/5), focusing on the technical merits of the cross-attention mechanism and the tethering loss. In contrast, reviewer VTwV assigns a borderline rating (3/5) with equal confidence (4/5), emphasizing significant concerns about clarity, methodology explanation, and empirical validation.
The central issue appears to be the paper's presentation rather than its technical foundations. While the approach itself shows promising results and innovative components, the explanation of these components and their implementation is insufficient. Key questions remain unanswered:

1. What is the precise nature of the tethering loss, and how does it differ from standard distillation techniques?
2. What distinguishes the contrastive loss from the alignment loss?
3. How exactly is MoCLIP integrated with other frameworks during training?
4. What evidence supports the claim that the resulting embeddings are more motion-aware?

However, aparts from the last concern, the others clarity issues are likely to be easily addressed in final revision for camera-ready.

---

### Decision · Program_Chairs · 2025-03-31

Accept